# Establishing a Consumer Quality Index for Fresh Plums (*Prunus salicina* Lindell)

**Carlos H. Crisosto**

Department of Plant Sciences, University of California, Davis, CA 95616, USA; chcrisosto@ucdavis.edu

**Abstract:** Plums are primarily marketed for fresh consumption, canning, freezing, jam and jelly. Unfortunately, plum consumption has remained steady or declined. Consumers complain about a lack of flavor quality but are willing to pay for higher quality. Thus, absence of flavor and cold storage disorders are the main barriers to consumption. Plum cultivars are sensitive to gel breakdown, flesh browning and 'off flavors'. Consumer approval and postharvest life are controlled by genotype, quality attributes, harvest date and proper postharvest handling. A consumer quality index (CQI) based on soluble solids concentration (SSC) and minimum firmness is proposed to maximize flavor and postharvest life. In most cases, late harvest increases quality attributes. Our work and industry experience demonstrated that using critical bruising thresholds (CBT) based on minimum firmness measured at harvest acts as a reliable predictor of how late to harvest safely for maximum visual and sensory quality. Plums tolerated late harvest well because of their high tolerance to bruising damage, but suitable postharvest temperature management and selling within the potential postharvest life are required to maintain flavor and avoid the beginning of chilling injury. Thus, to maximize flavor and postharvest life, a CQI based on SSC and minimum firmness measured at consumption is proposed. This article provides guidance on using this CQI, combined with proper postharvest handling techniques, such as correct harvest date determination and temperature management, to maintain quality and increase consumption.

**Keywords:** plum consumption; consumer quality index; flesh breakdown; temperature management; critical bruising thresholds; maximum maturity; late harvest; firmness; SSC





## 1. Introduction

Plum (*Prunus salicina* Lindell) is a fruit classified as a drupe, consisting of a single seed surrounded by a pericarp. The pericarp is differentiated into an outer skin, or exocarp, a fleshy middle layer, or mesocarp, and a hard, woody layer, or endocarp, surrounding the seed. The exocarp (skin) and mesocarp (flesh) are rich in carbohydrates (sugar alcohols and the soluble sugars sucrose, glucose and fructose), organic acids, fats, proteins, dietary fiber, minerals and vitamins. Nutrient concentrations have been reported for plums grown in the USA [1] and in Europe [2]. Plums are characterized by high concentrations of antioxidants and bio nutrients compared to other fruits [3–5]. All these fruit components have sensory, nutritional, and health-promoting qualities. The bioactive compounds can be categorized into phenolic compounds and tetraterpenoids that play a role in cellular metabolism and in cosmetic look (pigmentation and browning) and taste (astringency) of the fruit [6–9]. Phenolic compounds are secondary metabolites that can be divided into phenolic acids and flavonoids [4,5]. These pigments are the basis of antioxidants, vitamin C and carotenoids. Chlorogenic and caffeic acids are abundant as primary contributors to phenolic acids. Flavonoids are a large group of structurally related compounds that include anthocyanins, flavones, flavonols, flavanones, flavan-3-ols and isoflavones. Anthocyanins, one abundant flavonoid pigment, is responsible for the red or black color of plums.

Due to their high concentrations of phenolic acids, flavonoids and anthocyanins, eating plums reduces generation of reactive oxygen species (ROSs) in human blood plasma

and provides defense from several chronic diseases. In particular, plum fruit polyphenols have chemo-preventive properties against estrogen-independent and -dependent breast cancer cells, with small or no activity on normal cells [6,7], inhibition of growth and induction of differentiation of colon cancer cells [8], attenuation of oxidative stress and inflammation in in vitro and ex vivo studies, inhibition of tumor growth and metastasis of breast cancer in mice [9], prevention of peril factors for obesity-related metabolic disorders and cardiovascular disease in rats [7] and alteration of intestinal microbiota in rats [5–11]. Plums also have laxative and antihypertensive properties and are suitable for managing constipation and treating duodenal ulcers [5,11].

## 2. Results

### 2.1. Developing Critical Bruising Thresholds and Maximum Maturity Tools

To develop a safe protocol to decide how late an orchard can be harvested, I defined the maximum maturity index [12,13] as the latest stage at which plums can be harvested without suffering bruising damage during commercial postharvest handling [14,15]. For this, I developed maximum maturity indices for plum cultivars using bruising susceptibility measurements according to fruit firmness [13–17]. Bruising susceptibility was determined by subjecting fruit of different firmness to three industry-common bruising energy levels (G), measured with an IS-100 accelerometer (Techmark, E. Lansing, MI, USA). The three dropping heights onto a surface of known physical characteristics simulated the impact bruising energy G levels detected in our previous packinghouse bruising potential survey [16,18,19].

### 2.2. Fruit Quality Attributes and Consumer Acceptance

Plums undergo biochemical and physical ripening changes after reaching maturity [13,19]. Quality attributes include sweetness (carbohydrates) increase, sourness (organic acids) perception, firmness decrease and fruit color shifts from green to red or dark during ripening. Our previous work using one early-, one mid- and one late-season important cultivar grown at three locations indicated that in most plums, delaying harvest increased fruit size, intensified red color and improved flavor, increasing quality attributes attractive to consumers [17,20–22]. During this survey, orchards did not affect sensory attributes, but harvest date and cultivar did. However, some bruising and decay problems were observed during cold storage and retail handling, indicating that a non-destructive approach to determining how late harvest can be carried out was important ([15], Table 1).

Plum consumption has remained steady or even declined, while consumers complain of a lack of flavor quality. Thus, I decided to investigate these consumption barriers [14,20,22,23]. Consumer visual and sensory acceptance is mainly related to SSC (soluble solid concentration), TA (titratable acidity), SSC: TA, color, and firmness [9,15]. In plums, phenolic content (astringency) in the skin can cause consumers to reject some cultivars [8,20]. Trained panel and consumer acceptance sensory work [21,23] revealed that plum consumer acceptance reaches its maximum potential (80 to 90%) when fruit is consumed at a firmness of 0.9 to 1.8 kilos (ready-to-eat, too soft). If plums are consumed at a higher firmness (less ripe to hard, 1.8 to 3.6 kilos), consumer acceptance falls from ~85% down to ~40% [20]. SSC levels drive consumer acceptance of most plums, and TA becomes a factor on consumer decision mostly for early-season plums with high TA > 0.7%. In 'Blackamber' plums, SSC and TA levels were highly dependent on harvest date, thus consumer acceptance and market life were as well (20, Table 2). For plums within the most common industry ripe soluble solids concentration (RSSC) range (10.0 to 11.9%), ripe titratable acidity (RTA) played a significant role in consumer acceptance. Plums within this RSSC range combined with low RTA (≤0.60%) were disliked by 18% of consumers, while plums with RTA ≥ 1.00% were disliked by 60% of consumers. Plums with RSSC ≥ 12.0% had ~75% consumer acceptance, regardless of RTA (Table 2). Our sensory work also determined that ripening of harvested plums treated before consumption decreased TA by 30 to 40% from the TA measured at harvest. In some cases, this

decrease in TA and increase in the SSC:TA ratio during ripening may increase the acceptability of plums that would otherwise be unacceptable. Ripening protocols prior to delivery to retail stores, at retail stores and at home have been developed and promoted [18,20,24].

**Table 1.** Japanese plum fruit composition (based on 100 g raw halves).

| Name | Amount | Unit | Name | Amount | Unit |
|---|---|---|---|---|---|
| Water | 87.2 | g | Copper, Cu | 0.057 | mg |
| Energy | 46 | kcal | Selenium, Se | 0 | µg |
| Protein | 0.7 | g | Vitamin C, total ascorbic acid | 9.5 | mg |
| Total lipid (fat) | 0.28 | g | Thiamin | 0.028 | mg |
| Carbohydrate, by difference | 11.4 | g | Riboflavin | 0.026 | mg |
| Fiber, total dietary | 1.4 | g | Niacin | 0.417 | mg |
| Sugars, total including NLEA | 9.92 | g | Vitamin B-6 | 0.029 | mg |
| Calcium, Ca | 6 | mg | Folate, total | 5 | µg |
| Iron, Fe | 0.17 | mg | Folic acid | 0 | µg |
| Magnesium, Mg | 7 | mg | Folate, food | 5 | µg |
| Phosphorus, P | 16 | mg | Folate, DFE | 5 | µg |
| Potassium, K | 157 | mg | Choline, total | 1.9 | mg |
| Sodium, Na | 0 | mg | Vitamin B-12 | 0 | µg |
| Zinc, Zn | 0.1 | mg | Vitamin B-12, added | 0 | µg |
| Carotene, beta | 190 | µg | Vitamin A, RAE | 17 | µg |
| Carotene, alpha | 0 | µg | Retinol | 0 | µg |
| Cryptoxanthin, beta | 35 | µg | MUFA 16:1 | 0.002 | g |
| Lycopene | 0 | µg | MUFA 18:1 | 0.1 | g |
| Lutein + zeaxanthin | 73 | µg | MUFA 20:1 | 0 | g |
| Vitamin E (alpha-tocopherol) | 0.26 | mg | MUFA 22:1 | 0 | g |
| Vitamin E, added | 0 | mg | Fatty acids, total polyunsaturated | 0.044 | g |
| Vitamin D (D2 + D3) | 0 | µg | PUFA 18:2 | 0.044 | g |
| Vitamin K (phylloquinone) | 6.4 | µg | PUFA 18:3 | 0 | g |
| Fatty acids, total saturated | 0.017 | g | PUFA 18:4 | 0 | g |
| SFA 4:0 | 0 | g | PUFA 20:4 | 0 | g |
| SFA 6:0 | 0 | g | PUFA 20:5 n-3 (EPA) | 0 | g |
| SFA 8:0 | 0 | g | PUFA 22:5 n-3 (DPA) | 0 | g |
| SFA 10:0 | 0 | g | PUFA 22:6 n-3 (DHA) | 0 | g |
| SFA 12:0 | 0 | g | Cholesterol | 0 | mg |
| SFA 14:0 | 0 | g | Alcohol, ethyl | 0 | g |
| SFA 16:0 | 0.014 | g | Caffeine | 0 | mg |
| SFA 18:0 | 0.003 | g | Theobromine | 0 | mg |
| Fatty acids, total monounsaturated | 0.134 | g | | | |

https://fdc.nal.usda.gov/fdc-app.html#/food-details/169949/nutrients (accessed on 9 March 2023).

New cultivars released, especially pluots, were selected based on their low TA, high SSC, low astringency in the skin and high consumer acceptance potential [21,24].

### 2.3. Barriers to Plum Consumption

Cold Storage Disorders that Limit Consumption: Chilling injury symptoms (CI) such as gel breakdown, flesh translucency, flesh bleeding, flesh browning and/or 'off flavor' (Figure 1) normally appear in ripe fruit after warming at ripening temperatures (18 °C to 25 °C) following cold storage [25–28]. The beginning and intensity vary among cultivars and are greatly affected by temperature management and harvest maturity [25,27,28]. Most plums and pluots are more susceptible to expression of these symptoms when stored at 5 °C rather than 0 °C [28], (Table 3). Thus, market life, defined as when 20% of fruit show symptoms, varies by cultivar and storage temperature [28], (Table 3). For example, the market life of 'Blackamber', 'Fortune', 'Betty Ann', 'Joana Red', 'Flavorich, 'October Sun' and 'Angeleno' plums stored at 0 °C was >five weeks, while other tested cultivars developed chilling injury symptoms within three–four weeks, even when stored at 0 °C [28],

(Table 3). In all plum cultivars, a much longer market life was achieved after storage at 0 °C than at 5 °C [28], (Table 3). However, market-life potential is also influenced by other factors such as orchard conditions, season and maturity. Pit burning symptoms that look like flesh browning disorders are associated with high temperatures during fruit maturation (heat damage) and are triggered during the growing season. Thus, when determining market life, it is critical to assess fruit condition on ripe fruit to detect this field problem prior to placing fruit in cold storage.

**Table 2.** Quality attribute changes for three cultivars growing in two locations and harvested at three maturities.

| Cultivar | Loc-Mat-- | Date | Firm. (%) | S.S.C. (%) | T.A. (%) | SSC/A Ratio | Color | Phenol mg/100 mL | $CO_2$ Peak mg/kg·h | $C_2H_4$ Peak mg/kg·h |
|---|---|---|---|---|---|---|---|---|---|---|
| **Plums** | | | | | | | | | | |
| **Blackamber** | 1-1 | 6/20 | 8.0 | 11.1 | 1.21 | 9.2 | 7.1 | 48.2 | 35 | 0.3 |
| | 1-2 | 6/20 | 7.4 | 11.5 | 1.06 | 10.8 | 5.5 | 46.0 | 55 | 45 |
| | 1-3 | 6/20 | 6.1 | 12.2 | 0.71 | 17.2 | 2.5 | 40.5 | 69 | 115 |
| | 2-1 | 6/25 | 7.2 | 11.5 | 1.15 | 10.0 | 7.7 | 38.1 | 37 | 0.1 |
| | 2-2 | 6/25 | 6.4 | 11.5 | 0.81 | 14.2 | 5.3 | 34.5 | 35 | 0.2 |
| | 2-3 | 6/25 | 6.1 | 12.4 | 0.73 | 17.0 | 3.4 | 33.0 | 59 | 79 |
| **Friar** | 1-1 | 7/12 | 8.8 | 13.7 | 0.95 | 14.2 | 5.1 | 62.8 | 35 | 0.1 |
| | 1-2 | 7/12 | 8.4 | 14.8 | 0.78 | 19.0 | 3.5 | 57.4 | 39 | 0.8 |
| | 2-1 | 7/17 | 7.5 | 13.6 | 0.73 | 18.6 | 5.8 | 48.6 | 63 | 81 |
| | 2-2 | 7/17 | 6.7 | 15.3 | 0.61 | 25.1 | 2.8 | 60.6 | 64 | 98 |
| | 3-1 | 7/20 | 7.6 | 13.9 | 0.85 | 16.4 | 5.8 | 55.9 | 39 | 0.2 |
| | 3-2 | 7/20 | 7.4 | 15.4 | 0.72 | 21.4 | 3.7 | 54.8 | 59 | 76 |
| **Angeleno** | 1-1 | 9/5 | 9.8 | 17.8 | 0.48 | 37.1 | 9.7 | 59.4 | 26 | 0.1 |
| | 1-2 | 9/5 | 8.7 | 17.1 | 0.45 | 38.0 | 4.7 | 68.1 | 24 | 0.1 |
| | 2-1 | 9/14 | 8.2 | 17.8 | 0.48 | 37.1 | 5.8 | 57.7 | 28 | 0.1 |
| | 2-2 | 9/14 | 6.9 | 17.6 | 0.39 | 45.1 | 4.1 | 71.6 | 28 | 0.1 |

Maturity (Mat); Firm.: Flesh firmness, 5/16-inch tip; S.S.C.: soluble solids content; T.A.: titratable acidity. Color: peach and nectarine C.T.F.A. color chips, low number is greener. Plums: U.C.D. red color chips, low number is darker.

**Table 3.** Mature and ripe 'Blackamber' quality attribute changes at four harvest dates, adapted with permission from Ref. [20].

| | | | At Harvest | | | | | | After Ripening [Z] | | |
|---|---|---|---|---|---|---|---|---|---|---|---|
| | | | Flesh color | | | | | | | | |
| Harvest | Fruit mass (g) | Cheek firmness (N) | L | Chroma | Hue angle | SSC (%) | TA (%) | SSC: TA | SSC (%) | TA (%) | SSC: TA |
| 1 | 109.2 | 110 | 67.9 | 28.8 | 100.3 | 10.3 | 1.15 | 9.2 | 10.3 | 0.78 | 13.2 |
| 2 | 118.0 | 223 | 67.5 | 28.9 | 95.7 | 10.6 | 0.70 | 14.2 | 10.8 | 0.47 | 23.8 |
| 3 | 121.4 | 197 | 64.6 | 28.5 | 89.2 | 11.7 | 0.50 | 24.2 | 11.7 | 0.43 | 27.4 |
| 4 | 122.3 | 134 | 62.8 | 28.3 | 85.7 | 11.9 | 0.42 | 28.2 | 12.3 | 0.33 | 37.5 |
| | <0.0001 | <0.0001 | 0.0001 | 0.6640 | 0.0001 | 0.0001 | <0.0004 | <0.0022 | <0.0092 | 0.0001 | 0.0001 |
| | 4.1 [Y] | 2.22 | 1.4 | NS | 0.8 | 0.5 | 0.24 | 3.7 | 1.09 | 0.06 | 3.7 |

[Z] Fruit ripened at 20 °C and 85% RH until flesh firmness reached 8.9 to 17.8 N. [Y] Mean separation by LSD test at *p* > 0.05.

### 2.4. Preventing Cold Storage Injury

Greater potential postharvest life was achieved when stored at 0 °C than at 5 °C for most plum cultivars [28], (Table 4). In addition, late harvest maturity can also extend postharvest life [15,27–29]. Field heat should be removed using field bins, forced-air cooling, hydro-cooling or room cooling immediately after harvest prior to packing. Then, packed plums should be cooled by forced-air cooling to near 0 °C. A storage temperature of −1.1 °C to 0 °C with 85 to 95% RH is highly recommended to maximize plum postharvest life. However, to store plums at this low a temperature of −1.1 °C, high SSC and excellent

thermostatic control are essential to avoid freeze damage [30]. Plum freezing point varies from −2 to −1 °C, depending on SSC [30], (Table 5).

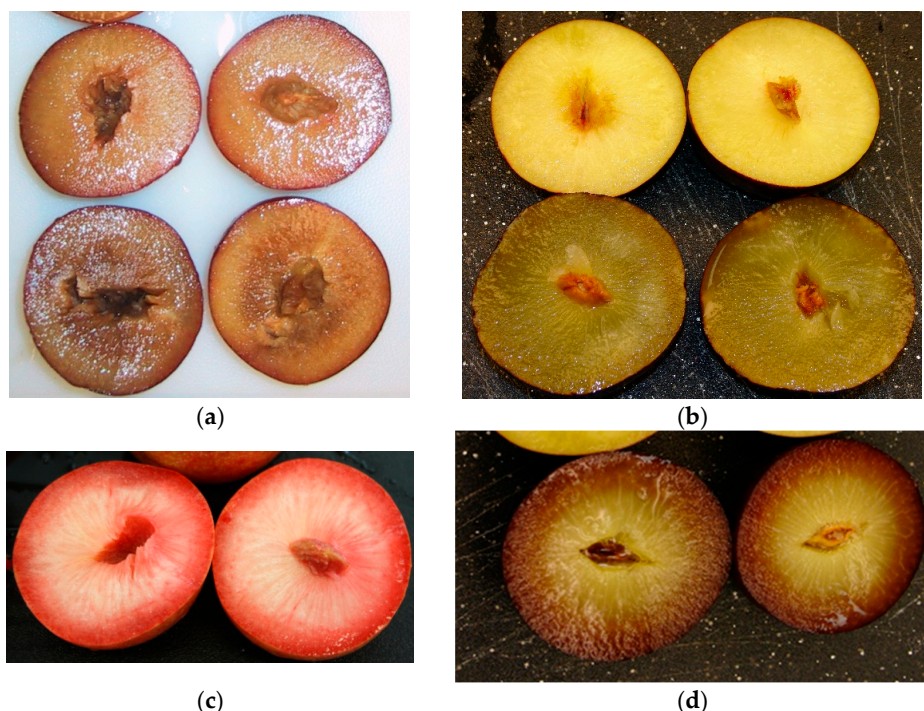

(**a**)  (**b**)

(**c**)  (**d**)

**Figure 1.** Plum chilling injury symptoms observed during cold storage: gel breakdown, flesh browning, flesh bleeding, flesh translucency and overripe. (**a**) Plum showing fresh browning and gel breakdown overripe symptoms. (**b**) Sound plums on top and plum showing gel breakdown and translucency on the bottom. (**c**) Plum showing flesh bleeding. (**d**) Plum showing overripe symptoms. Photos courtesy of Dr. Carlos H. Crisosto (plum diameters vary from 6 to 8 cm).

**Table 4.** Storage Temperature and duration on Market Life Potential of Plum Cultivars.

| Storage/Shipping Potential (Weeks) | | | | |
|---|---|---|---|---|
| Cultivar | Plant breeding program | Fruit type | 0 °C | 5 °C |
| Angeleno | Garabedian | Semi-free to freestone | 5 | 5 |
| Betty Anne | Zaiger | Clingstone | 5 | 5 |
| Blackamber | Weinberger | Freestone | | |
| Earliqueen | Zaiger | Clingstone | 3 | 2 |
| Friar | Weinberger | Freestone | 5 | 3 |
| Flavorich | Zaiger | Clingstone | 5 | 5 |
| Fortune | Weinberger | Semi-clingstone | 5 | 3 |
| Hiromi Red | Zaiger | Clingstone | 5 | 3 |
| Howard Sun | Chamberlin | Freestone | 4 | 1 |
| Joanna Red | Zaiger | Freestone | 5 | 5 |
| October Sun | Chamberlin | Semi-clingstone | 5 | 5 |
| Purple Majesty | Bradford | Clingstone | 5 | 3 |
| Showtime | Wuhl | Freestone | 5 | 3 |

Controlled atmosphere (CA) and/or modified atmosphere packaging (MAP) can retain fruit firmness and delay changes in ground color and decay in some cases. Since commercial outcomes have been erratic and even damaging, the reliable benefits provided by low temperature storage, CA and MAP have limited commercial use [31,32].

**Table 5.** Relationship between Soluble Solids Content (SSC) and the Freezing Point.

| SSC | | Safe Freezing Point |
|---|---|---|
| (%) | (°F) | (°C) |
| 8.0 | 30.7 | −0.7 |
| 10.0 | 30.3 | −0.9 |
| 12.0 | 29.7 | −1.3 |
| 14.0 | 29.4 | −1.4 |
| 16.0 | 28.8 | −1.8 |
| 18.0 | 28.5 | −1.9 |

*2.5. The Lack of Plum Consumer Quality Index (CQI)*

Establishing a Consumer Quality Index

Harvest date is determined by skin background color changes (green to dark or red) using a color chip guide to assess harvest maturity [14–17,19]. In new cultivars, full red or black color develops prior to harvest maturity, masking background color changes on the fruit skin and making this maturity index impractical. Using background color changes as a harvest maturity index only guarantees that in most cases, plums will ripen off the tree. However, this harvest maturity index does not assure that plums will reach minimum quality attributes to satisfy consumers or reach their maximum postharvest life. Therefore, to increase plum consumption, a consumer quality index (CQI) concept was needed. A consumer quality harvest index was developed based on sensory work and critical bruising thresholds (CBT) that describe fruit susceptibility to physical abuse. Based on 'in-store' consumer test results, a minimum SSC of 11 to 12% for selected plum cultivars will satisfy at least 85% of consumers. Furthermore, our plum-trained panel segregated cultivars according to the perception of sensory characteristics by consistently allocating plum cultivars into tart, plum aroma, and sweet/plum flavor groups [21]. For each flavor group, SSC was the main driver of consumer acceptance [21,23], except for cultivars in the tart group, when acidity was at least 0.7%.

**3. Discussion**

*3.1. CBTs*

Plums subjected to these energy bruising levels developed a low number of bruises at the high energy level 245 G (very rare) only on fruit below 1.4 kilos [12], (Table 6). The use of the CBTs allows us to harvest later without causing mechanical damage, thereby maximizing potential fruit quality. Under specific conditions, the assessment of fruit damage susceptibility and packing line G-forces will help determine how late and how soft fruit can be harvested and packed without causing bruising. Maximizing fruit quality potential depends on the cultivar and/or orchard conditions.

**Table 6.** Minimum flesh firmness measured at the weakest point on the fruit necessary to avoid commercial bruising at three levels of physical handling (CBT, critical bruising threshold index) expressed as height (cm) or G acceleration forces.

| | Drop Height [z] | | | |
|---|---|---|---|---|
| | (1.0 cm) | (5.1 cm) | (10.2 cm) | Weakest |
| Cultivar | ~66 g | ~185 g | ~246 g | position |
| Blackamber | 0 | 0 | 3 [y] | Tip |
| Fortune | 0 | 0 | 0 | Shoulder |
| Royal Diamond | 0 | 0 | 0 | Shoulder |
| Angeleno | 0 | 0 | 0 | Shoulder |

[z] Dropped onto a 0.3 cm PVC belt. Damaged areas with a diameter equal to or greater than 2.5 mm were measured as bruises. [y] Fruit firmness was measured with an eight mm tip penetrometer.

Our previous stone fruit transportation work indicated that packaging system and fruit firmness cause damage during transportation (Crisosto's unpublished data). Tray-packed stone fruit stands transportation better than volume-filled. Fruit with firmness of 2.3 to 3.6 kilos on the weakest fruit position had up to 2% potential damage.

At retail, bruising potential was measured by placing an IS-100 recording accelerometer in the center of the top layer of 12 two-layer tray-packed boxes (45 cm × 36 cm × 16.5 cm box size, 48 fruit), which indicated that accelerations (G) and velocity changes (m/s) varied during box handling, removal from the pallet and repalletization (12, Table 7). At this step, the force of impact varied from 32 G (7.6 cm drop) to 103 G (31 cm drop). Accelerations during box handling and retail display ranged from 19.6 to 34.7 G. Thus, accelerations at the retail level were lower than the CBT for many plums with firmness equal to or greater than 1.4 kg force.

**Table 7.** Acceleration and velocity change measured in the center position of the top tray of two-layer, tray-packed metric boxes dropped from different heights (cm) onto a solid countertop. Values given are means (±standard deviations).

| Drop Height (cm) | Acceleration (G) | Velocity Change (m/s) |
|:---:|:---:|:---:|
| 7.6 | 32.4 (±3.0) | 1.10 (±0.02) |
| 15.2 | 58.2 (±4.2) | 1.64 (±0.06) |
| 22.9 | 89.1 (±2.7) | 2.18 (±0.13) |
| 30.1 | 103.6 (±11.6) | 2.54 (±0.21) |

In most cases, using these CBTs to determine harvest date allowed SSC to accumulate and exceed the proposed consumer quality index (CQI). This protocol maximizes the number of plums in the orchard, exceeding the proposed CQI in most orchards (Table 8).

**Table 8.** Proposed consumer quality indexes (CQIs) based on two key components: firmness (eight mm tip) and minimum SSC for different plum cultivars measured at harvest.

| Cultivar | Firmness (Newtons) | Minimum SSC (%) |
|:---|:---:|:---:|
| Blackamber | 31.2–40.1 | 10–12 [Z] |
| Fortune | 31.2–40.1 | 11 |
| Friar | 31.2–40.1 | 11 |
| Royal Diamond | 31.2–40.1 | 11 |
| Angeleno | 26.7–40.1 | 12 |
| Betty Anne | 31.2–40.1 | 12 |

[Z] Plums with TA ≤ 0.60% after ripening have high consumer acceptance; however, if plums have ≥12.0% SSC, TA does not play a role.

### 3.2. Harvesting and Handling Plums

In California, harvest date is determined by skin color changes as described for each cultivar [14]. A color chip guide is used to determine maturity for some cultivars. Firmness, measured by squeezing fruit in the palm of the hand ('spring'), is also a useful maturity index for a few cultivars [14,33], especially those that achieve full color several weeks prior to harvest. Measurement of fruit firmness is recommended for plum cultivars where skin ground color is masked by full red or dark color development before maturation. Flesh firmness, measured using a penetrometer (eight mm tip), can be used to determine the proposed CBT and maximum maturity index, which is the stage at which fruit can be harvested without suffering bruising damage during postharvest handling. Plums are less susceptible to bruising than most peach and nectarine cultivars at comparable firmness.

### 3.3. Application of CBTs—Consumer Quality Index in the Field

Plum ripening on the tree progresses from the top of the tree to the bottom, a consequence of temperature exposure, light environment, and canopy structure. Lower fruit can mature as much as 10 to 14 days later than well-exposed fruit at the top of the tree.

Consequently, multiple harvests are conducted, generally two to four. The first harvest of plums is commonly the largest pick. Since many plum cultivars develop full color up to several weeks before commercial harvest and usually soften relatively slowly, it was important to develop a method by which a field laborer can easily determine fruit maturity and predict consumer quality potential. In such full-color cultivars, this is commonly achieved by limiting harvest to only a portion of the tree, usually segregated by light exposure: the top third of the tree in the first harvest, the middle third in the second, and so on, so that workers can proceed more quickly. Fruits are harvested into picking bags that can hold up to ~20 kg of fruit. The pickers dump the fruit into bulk bins that contain ~400 to 450 kg of fruit. The bulk bins are transported within the orchard on tractor-pulled trailers that hold three or four bins. Two tractors and bin-trailers are usually required for each harvest crew. When full, the bins are taken to a centralized shaded area and unloaded from the bin-trailers to await loading by forklift onto flatbed trailers for delivery to the packing facility.

## 4. Conclusions

I propose using a CQI to increase plum consumption. To support the establishment of this CQI, I developed maximum maturity indices based on CBT for the most important plum cultivars, using bruising susceptibility measurements based on fruit energy impact firmness. These CBTs were calculated for different levels of fruit firmness and expressed as G (acceleration). These thresholds predict how much physical abuse fruit will tolerate at different firmness levels during packinghouse operations. The use of CBT allows later harvesting without inducing bruising, maximizing fruit quality attributes and consumer acceptance. Most plum cultivars do not develop physical damage when exposed to high energy and/or bruising impact, thus, late harvest can be practiced. Harvesting late, in most cases, allows SSCs that exceed the CQIs to be achieved without jeopardizing the crop. I propose using CBTs as a tool to assist harvest decisions without inducing bruising during postharvest handling, thereby maximizing fruit orchard quality potential. Other factors on when to harvest should also considered such as fruit drop, environmental conditions, hand labor availability, market prices, distance to market, potential transportation damage and temperature management at the receiving location.

A controlled late-maturity harvest will allow fruit to gain sensory attributes and reduce cold storage damage, overcoming two primary plum consumption barriers. The establishment of this CQI combined with proper temperature management should ensure highly flavored fruit for consumers.

**Funding:** This work was funded by the Department of Plant sciences, University of California Agriculture and Natural Resources.

**Data Availability Statement:** Data is available upon request to the author.

**Acknowledgments:** The author would like to recognize Michael Thurlow, Summeripe Worldwide Inc. Gayle Crisosto, David Garner, and George Manganaris for their help during these studies.

**Conflicts of Interest:** The author declares that he has no known competing financial interest or personal relationship that could have appeared to influence the work reported in this paper.

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
