# Peer review of "Establishing a Consumer Quality Index for Fresh Plums (Prunus salicina Lindell)"

_horticulturae, doi:10.3390/horticulturae9060682_

Round 1
Reviewer 1 Report
In this manuscript (horticulturae-2420275) entitled ‘Establishing a Consumer Quality Index for Fresh Plums (Prunus salicina Lindell)’ submitted to Horticulturae, Carlos Haroldo Crisosto has proposed a consumer quality index (CQI) based on soluble solids concentration (SSC) and minimum firmness to maximize flavor and postharvest life. This manuscript provides guidance on using this CQI, combined with proper postharvest handling techniques such as correct harvest date determination and temperature management, to maintain quality and increase consumption. Overall, I consider this review topic interesting and complete, but some minor issues need to be addressed for improving the quality of this manuscript.
1, I am confused about the article type of this manuscript. Although it is labelled as review paper, this manuscript contains sections appeared in research article, including methods and materials, results, and discussion… Please clarify in the revised manuscript.
2, A figure to show the key component for establishing the consumer quality index (CQI) should be provided in the revised manuscript.
3, For the Figure 1, the scale bar should be included in the revision.
4, Please consider to cite the original papers instead of citing other reviews or book chapters when possible, especially for reference 16 to reference 37. The authors should direct or recommend the readers to consult original reference in case they wish to go into detail.
5, Since Dr. Carlos H. Crisosto is the only contributer to this manuscript, 'I' instead of 'We' are proposed to be employed in this maintext.
Author Response
Reviewer 1
Thanks for your excellent comments:
1, I am confused about the article type of this manuscript. Although it is labelled as review paper, this manuscript contains sections appeared in research article, including methods and materials, results, and discussion… Please clarify in the revised manuscript.
That was my mistake, I was confused too when I started to write it, but after discussing with editors, this is a review for their Physiological Characteristics and Postharvest Quality of Fruit that included old and new information on plum quality to tell the full story and some recommendations. I removed the M&M part to avoid confusing.
2, A figure to show the key component for establishing the consumer quality index (CQI) should be provided in the revised manuscript.
I made changes in the Table 2 to include your comment as we are getting too long.
3, For the Figure 1, the scale bar should be included in the revision. At this point, I will deal with that mentioning the commercial plum diameter. Next time I should put the ruler when I take the photo. These are old photos from 1993 when we were doing a lot of work in plums.
4, Please consider citing the original papers instead of citing other reviews or book chapters when possible, especially for reference 16 to reference 37. The authors should direct or recommend the readers to consult original reference in case they wish to go into detail.
Good improvement I did! And removed 2 -3 review papers as were previously mentioned in the text.
5, Since Dr. Carlos H. Crisosto is the only contributor to this manuscript, 'I' instead of 'We' are proposed to be employed in this main text.
I understand but the full work was done with my group and visitors over years so in some case I mentioned as “We” and in cases when I propose I add I. I will add some of them in the knowledge statement.
Reviewer 2 Report
Dear author,
The article entitled "Establishing a Consumer Quality Index for Fresh Plums 2 (Prunus salicina Lindell)" is interesting and provide a new paradigm in this particluar field. However, it is not clear that this article is categorized as review article or original research article. If it is a review article, please provide the references/sources for each data shown in this study. If it is an original research article, please write the method of analysis in more detail.
Author Response
Reviewer 2
I improved organization as I was confused between research and review format when I started to write it, but after discussing with editors, this is a review for their Physiological Characteristics and Postharvest Quality of Fruit that included old and new information on plum quality to tell the full story and some recommendations. I removed the M&M part to avoid confusing and improve organization.
Reviewer 3 Report
It is a manuscript titled "Establishing a Consumer Quality Index for Fresh Plums (Prunus salicina Lindell)"; in which the background is focused on the characterization of bioactive compounds and their impact on some health benefits (cancer, ROSs, oxidative stress, obesity, among others), but at no time does it address issues about the parameters that are used to evaluate quality for consumers. In addition to writing the manuscript, despite being a review, it has a research article component by proposing a section on material and methods, which lacks the fundamental steps of the scientific method since it does not have an objective or hypothesis. The material and methods section does not present clear details of the experimental design, how many experimental units were tested for each effect, it is necessary to describe the methodologies in detail so that the research could be replicated. Nor does it present a proposal for statistical analysis of the data obtained.
In the results section, it alludes to and describes the results of other investigations with results that have presumably already been published, but none of them cites the source to know the details of the origin of that information.
It presents a lot of information without having any reference to support it and it is unknown how that information was obtained.
In general, it appears to be an autobiography of articles published by the author and other contributors, but to be an attractive review, most of the information should be from the last 3 years, and in this case, the highest proportion of citations it has are from more than ten years old.
Additionally, I make some specific observations that should be considered:
L14: "Our work", the writing of a manuscript must be done in the third person, please review the entire document and change the wording.
L37-39: There must be bibliographic support for this information.
L41-45: Likewise, each paragraph must have scientific support that confirms or supports the information presented.
L46-48: You must support that information.
L62: "we defined" If there is only one author of this manuscript, why does he speak in the plural?
L64; "We developed" the same comment as the previous one, because plural if it is a single author.
L67: "fruit with different firmness" how did you determine the different firmness? Was it subjective or instrumental? And from what values did you consider these firmness differences, and if they are published data, why don't you cite the source?
L68-69: "The three dropping heights onto a surface of known characteristics" should describe each of the dropping heights. Also describe the characteristics of the surface, since for me, as for many readers, they will be unknown.
L70: "detected in our previous packinghouse bruising potential survey" It speaks of information completely unknown to readers and does not refer to which publications it refers to.
L73-76; This paragraph is not a result of this study, that information should go in the introduction section.
L76-83: "Our previous work using one early, one mid and one late-season important cultivar grown at three locations..." Excuse me, but I don't know your previous works, let alone what they are about, since He doesn't even put the appointment to which he refers so as to be able to consult them.
L82-83: "(Table 2)." The information presented in Table 2 are not results of this work, so it is out of place. If you want to consider them as part of this manuscript, you must describe all the procedures used for each variable studied in the section on material and methods or if they are already published data, you must cite the source from which they are extracted.
L86: The meaning of the initials 'TA' is not described, you must describe the meaning when they are used for the first time.
L95, 100: "(Table 3)" if they are data published in other articles, you must cite the source; If they are unpublished data, you must describe the evaluation conditions in the material and methods section.
L196-200: "Our recent work on transportation-derived stone fruit..." Too many paragraphs that talk about our work, without even putting the quote to know which one it is. The author assumes that all readers already know the full range of publications that he has.
L216-218: They do not describe the number of boxes used for each variable and if they used descriptive statistics to express the results, it should be described in the material and methods section.
Author Response
Reviewer 3
I apologize for creating confusion, that was my mistake, I was confused too when I started to write it, but after discussing with editors, this is a review for their Physiological Characteristics and Postharvest Quality of Fruit that included old and new information on plum quality to tell the full story and some recommendations.
I removed the M&M and changed format a little it to avoid confusing.
it is a manuscript titled "Establishing a Consumer Quality Index for Fresh Plums (Prunus salicina Lindell)"; in which the background is focused on the characterization of bioactive compounds and their impact on some health benefits (cancer, ROSs, oxidative stress, obesity, among others), but at no time does it address issues about the parameters that are used to evaluate quality for consumers. In addition to writing the manuscript, despite being a review, it has a research article component by proposing a section on material and methods, which lacks the fundamental steps of the scientific method since it does not have an objective or hypothesis.
The material and methods section does not present clear details of the experimental design, how many experimental units were tested for each effect, it is necessary to describe the methodologies in detail so that the research could be replicated. Nor does it present a proposal for statistical analysis of the data obtained.
In the results section, it alludes to and describes the results of other investigations with results that have presumably already been published, but none of them cites the source to know the details of the origin of that information. It presents a lot of information without having any reference to support it and it is unknown how that information was obtained.
I added citations in the text to support my statements.
In general, it appears to be an autobiography of articles published by the author and other contributors, but to be an attractive review, most of the information should be from the last 3 years, and in this case, the highest proportion of citations it has are from more than ten years old.
This is a review that combined interesting old data and new data to tell the full story on how to overcome consumption barrier in Japanese plums. I agree that I am using very old data but reliable and interesting (late 1980s) to justify my approach to improve consumer quality. keep on mind that CA produces almost 85% of Japanese plum in US so most of research information is produce in the state. This old data has not been very limited to public and now as part of this review will become public to our peers. Probably this will be the last time that peers will have the opportunity to access to this data as most of the plum researchers are dead or leaving the system.
Additionally, I make some specific observations that should be considered:
L14: "Our work", the writing of a manuscript must be done in the third person, please review the entire document and change the wording.
I went over these comments, as this review is based on work carried out by my group and/or visitors I used We sometimes when I am proposing something as my opinion, I used I. I made the changes and also add some important members of my group and visitors names in the knowledge section.
L37-39: There must be bibliographic support for this information. Added
L41-45: Likewise, each paragraph must have scientific support that confirms or supports the information presented. I added as was pertinent.
L46-48: You must support that information. I did
L62: "we defined" If there is only one author of this manuscript, why does he speak in the plural? OK
L64; "We developed" the same comment as the previous one, because plural if it is a single author. OK
L67: "fruit with different firmness" how did you determine the different firmness? Was it subjective or instrumental? And from what values did you consider these firmness differences, and if they are published data, why don't you cite the source? I added the reference (Crisosto, C.H.; Slaughter, D.; Garner, D.; Boyd, J. Stone fruit critical bruising thresholds. J Am Pomol Soc 2001, 55, 76–81; Valero, C.; Crisosto, C.H.; Slaughter, D. Relationship between nondestructive firmness measurement and commercially important ripening fruit stages for peaches, nectarines, and plums. Postharvest Biol Technol 2007, 44, 248–53).
L68-69: "The three dropping heights onto a surface of known characteristics" should describe each of the dropping heights. Also describe the characteristics of the surface, since for me, as for many readers, they will be unknown. Yes, surface physical properties are described in the reports and paper Crisosto, C.H.; Slaughter, D.; Garner, D.; Boyd, J. Stone fruit critical bruising thresholds. J Am Pomol Soc 2001, 55, 76–81.
L70: "detected in our previous packinghouse bruising potential survey" It speaks of information completely unknown to readers and does not refer to which publications it refers to. I added the ref (Crisosto, C.H.; Slaughter, D.; Garner, D.; Boyd, J. Stone fruit critical bruising thresholds. J Am Pomol Soc 2001, 55, 76–81)
L73-76; This paragraph is not a result of this study, that information should go in the introduction section. OK
L76-83: "Our previous work using one early, one mid and one late-season important cultivar grown at three locations..." Excuse me, but I don't know your previous works, let alone what they are about, since He doesn't even put the appointment to which he refers so as to be able to consult them. Paper and report references were added
L82-83: "(Table 2)." The information presented in Table 2 are not results of this work, so it is out of place. If you want to consider them as part of this manuscript, you must describe all the procedures used for each variable studied in the section on material and methods or if they are already published data, you must cite the source from which they are extracted. Reference added at the bottom.
L86: The meaning of the initials 'TA' is not described, you must describe the meaning when they are used for the first time. I spelled out in the text.
L95, 100: "(Table 3)" if they are data published in other articles, you must cite the source; If they are unpublished data, you must describe the evaluation conditions in the material and methods section. OK
L196-200: "Our recent work on transportation-derived stone fruit..." Too many paragraphs that talk about our work, without even putting the quote to know which one it is. The author assumes that all readers already know the full range of publications that he has. (References added, some of this work results are not being published)
L216-218: They do not describe the number of boxes used for each variable and if they used descriptive statistics to express the results, it should be described in the material and methods section. OK added.
Round 2
Reviewer 3 Report
The manuscript was considerably improved with the corrections made, the information it is presenting is now clear; You should only check the journal's rules to cite the sources of the tables.
I insist that the information should be updated with references from the last five years for a review article. But if the editor considers its publication pertinent with the corrections made, there is no problem for me.